# Deprescribing: Fashion Accessory or Fig Leaf?

**DOI:** 10.3390/pharmacy7020049

**Published:** 2019-05-23

**Authors:** Arnold G. Zermansky

**Affiliations:** School of Healthcare, University of Leeds, Leeds LS2 9UT, UK; gpjagz@leeds.ac.uk

**Keywords:** deprescribing, medication review, drug review, drug administration, drug management, prescribing

## Abstract

Deprescribing is the general practice fashion accessory that no prescriber can be seen without. However, it is in danger of becoming a “fig leaf” substitute for the entire medication review suite.

## 1. Background

Stopping redundant medication is not new. Thinking prescribers have been doing it forever. However, suddenly it has become fashionable. The patient benefits from risk reduction and a simpler treatment regimen. The prescriber benefits from reduced prescribing costs and fewer prescriptions to sign. The clinical pharmacist feels good about stopping a medicine that might harm the patient. The health service benefits from reduction in treatment costs. Pills that are not useful risk doing harm. Stopping over-prescription is a win for everyone.

There is burgeoning literature on deprescribing, including an interesting review by Reeve et al. that quotes over 100 references [1]. Even these are selective, not least because not all deprescribing is recorded under that name. But hard evidence of value is still lacking and equivocal. Reeve et al.’s paper is insightful, however, and the description of the process and the problems that may be encountered and how to avoid them are valuable, though much of the described method extends further than the term implies, and beyond their own definition of deprescribing. This is confusing, because what Reeves et al. describe in detail as “deprescribing” is a rather more complete medication review than merely stopping tablets. There is nothing wrong with this, of course, but it may confuse some readers. Words are important, and people generally believe they mean what they say. So, the unwary will assume that deprescribing means exactly that, and not the more holistic process Reeve et al. describe.

Of course, it always has been easier to continue to prescribe that to deprescribe. I was an early enthusiast for deprescribing, though I called it un-prescribing back in the 1980s when our practice was the lowest prescribing practice in Leeds and I used to irritate my partners by insisting on patients coming in at intervals to have their medicines reviewed [2].

I postulated that repeat prescriptions had inbuilt **therapeutic momentum**. Prescriptions (like to Newton’s planets) would continue forever in a straight line unless a force intervened to alter their trajectory.

Stopping a medicine is a step into the unknown, as much as is starting one. However, when starting a medicine there is generally a clear indication. There is then a well-trodden path from initial prescription to repeat, though sometimes it is a jump rather than a measured tread. Some prescriptions are always intended to be long term, such as hypotensives, statins, and antiepileptics. Others, such as analgesics, may be started to address a short term indication but are continued because the symptoms, or the symptoms’ owner, require it. There are other drugs that are appropriately prescribed for prolonged periods, but with the intention of stopping when the disorder has been resolved. These include antidepressants and oral steroids. The common ground of all of these is that the prescriber should ensure that the patient is benefitting from treatment before making the drug a repeat. This sounds self-evident, but sadly it is a stage that commonly goes by default, as we shall see.

There are, conversely, some drugs that it would be unthinkable to stop. These include medicines that replace a hormone (such as insulin or thyroxine). With most drugs, however, you can only tell if the drug is still needed if you withdraw it. This does not necessarily mean that every patient should have every drug withdrawn at intervals just to see if is doing any good. As with all medical interventions, it is important to balance risks against benefits. It has been shown that about 40% of patients are able to withdraw hypotensive medication without adverse consequences [3]. But the process of doing so in people who are taking two or three medicines, each of which has been carefully titrated, is labour-intensive for patient and physician and so disappointing for the 60% who have to be re-titrated that it may not be worth the effort. Withdrawal has its theoretical risks, though the papers with long follow-up that have quantified risk suggest that it is small [4].

The low-hanging fruit of deprescribing are those preparations that have little evidenced benefit. Therefore, stopping benzodiazepines or older sedatives seems an obvious choice. As with all deprescribing, this is not as simple as it sounds, because withdrawal effects are common, even when the dose is tapered very gradually. Most oral NSAIDs (non-steroidal Anti-inflammatory drugs) can be stopped, except in those with chronic inflammatory disease. The STOPP/START criteria are mainly based on this approach [5]. Older people often take medicines for which the indication is lost in the mists of telegraphic-style handwritten clinical records many decades old. Indeed, I picture older prescription items as archaeological. The drug itself tells you something about when it was initiated, though not necessarily what for. Medicines have fashions as everything else does, and many drugs were modish for a limited period until succeeded by something (that seemed) better. For example, few would now initiate enalapril for hypertension, or digoxin for heart failure.

## 2. The Crossword

The good thing about long-term medication is that the patient has got used to it. Even if it did cause an adverse effect in 1975 when it was started, the very fact that the patient still takes it suggests that it is not causing a noticeable adverse effect now—or so you might imagine. However, some adverse effects are only perceived on withdrawal:
A retired science professor reported that he had forgotten to pack his beta blockers (for hypertension) on his short break holiday. He found to his delight that he could do The Times crossword again. He had imagined that his age had fogged his thinking, but it was his supposedly non-lipid soluble beta blocker, that is meant not to cross the blood brain barrier.

## 3. The Risk of Stopping Medication

In spite of the above, most long term treatments were initiated for a good reason (or so it seemed at the time), and stopping the pills may, therefore, expose the patient to risk. The patient on anti-hypertensives may have an increase in blood pressure, or conceivably even a stroke. The patient on antidepressants may have a relapse. The patient on a proton pump inhibitor may have an exacerbation of dyspepsia or even gastric bleeding. 

Even if the withdrawal of the drug makes pharmacological sense, there may also be temporary drug withdrawal effects that make the process of withdrawal painful. This is especially true of antidepressants and benzodiazepines, but can occur with almost any drug, and if an incidental event occurs that happens to coincide in time with the drug withdrawal, the patient is likely to blame the withdrawal, even if it is pharmacologically improbable.

Withdrawal effects, even if not threatening, are not trivial. Although the drug can always be restarted, the fact that this happens will not endear the deprescriber to the patient or carer. 

## 4. Steely Repeats

People become attached to their medicines, quite often for reasons that have nothing to do with pharmacology. In the first ever study of repeat prescriptions in 1970, Balint suggested that “repeat prescriptions are written in steel and concrete and are not easily dismantled or remodelled” [6]. He was not referring to pharmacological drug dependence, but the fact that a repeat drug can be a tangible and comforting symbol for a long term doctor-patient relationship. Withdrawing the drug leaves the patient bereft and without visible means of support. It is common experience that patients’ eyes glaze over while they reminisce about “my doctor” who initiated the medicine in question. Quite often the doctor referred to has been retired or even dead for years. The implication that today’s consulting doctor is not (and perhaps never could be) “my doctor” can be deflating to the conscientious practitioner who has cared meticulously for the patient over the last decade. However, the drug in question is the last link between the patient and their trusted historic doctor. Break that link is at your peril!

Dickinson et al. explored the use of long-term antidepressants in an elderly population [7]. They report “Barriers to discontinuation are significant. Feelings of pessimism, negative associations with ageing, deteriorating health and fear of relapse all reinforce a patient’s desire to continue […].”

There are, of course, more tangible reasons why a patient may not want to stop a long term medicine. After all, it was prescribed for a reason, and the patient may (reasonably) fear that stopping it may lead to a recurrence of previous symptoms and disease. Furthermore, patients live in the current political climate and are aware of the financial problems of the National Health Service (NHS) in Britain. They are often suspicious of the motives of a health care professional who wants them to stop receiving a treatment that costs the NHS money. This is particularly relevant when the professional is not their regular prescriber, but an unknown person “parachuted in”. NHS drugs are provided free of charge to older and several other categories of patient. Others pay a charge per item dispensed. This means that patients are shielded from the full cost of their medicines. In countries where patients pay the full cost of their drugs, this can influence their adherence and their attitude to long term treatment, especially when the drugs are expensive or the patient struggles to pay. On the other hand, knowing you have paid hard-earned cash for your tablets may give you a stronger incentive to take them.

## 5. The Manner and Manners of Deprescribing

It is vital to consider the manner and manners of deprescribing. The consultation should be face-to-face and the clinician should have read the patient’s relevant records. There needs to be a reconciliation of the drug list, so that the clinical records, the MAR (Medicine Administration Records, used by care homes in Britain to record whether and when each dose of a medicine has been given), and the patient and carer’s perception of what is currently being taken match. There needs to be a sharing of patient ideas and fears, including an understanding of the patient’s perceptions of the reason for taking the drug. Where appropriate a clinical measurement (such as pulse rate, blood pressure, weight, blood monitoring) should be done, before and after the drug withdrawal. Withdrawal may need to be gradual with some drugs (even this is not easy, as many drugs do not come is tablet sizes that facilitate slow withdrawal). The patient needs to be sure that they can report any hitch. Finally, both patient and deprescriber need to have an agreed fall-back position if the withdrawal is unsuccessful. The process will generally need at least two consultations. An agreement not to stop a medicine, or a decision to restart it afterwards, should not be seen as a failure of the intervention.

There is also a strategic issue about consistency of approach between those caring for the patients. There should be a practice-wide policy about deprescribing, and this may need to be extended to people outside the practice where care is shared. Patients who are cared for, including those in homes, particularly need this broader approach.

All of these issues make de-prescribing more complex that it seems. The benefit/risk balance of the process is not easy to assess for any single intervention, let alone a package of interventions.

## 6. Deprescribing Plus

Not all pills that are stopped are redundant. Some are inefficient, ineffective, or cause harm or potential harm. This may indicate a swap of tablets rather than a simple stop. It also may mean that the patient ends up on just as many, or even more tablets. The new ones may even be more expensive. Therefore, the success of the process cannot be measured by a tablet count, or even a cost comparison. Success should be measured globally in counting the number of patients whose intervention is complete and whose illness control is adequate and stable, without adverse drug effects. This needs to be done in the context of patient (and carer) satisfaction with the process.

## 7. Pseudo-Deprescribing

Patients on tablets do not always take them. Some are selective about which they take; they take the ones that seem important: “I always take my heart pills, doctor.” That actually means “I take the ones that my friend Jane says *she* takes for her heart”—even if they are not for her heart at all. However, they do not take the ones that seem less important, especially if they give them socially unacceptable symptoms, like urinary frequency or wind. Some deliberately leave their tablets if they are going to the pub, for fear that they may interact with alcohol. Some people simply forget entirely, forget the dose that is meant to be taken in the middle of the day when they are busy, or just muddle them up.

Others juggle with doses to achieve the desired effect. Sometimes this seems sensible, such as delaying your diuretic dose until you have been shopping. Increasing the dose of analgesics when pain is bad sounds reasonable—unless the patient is escalating opiates. Some are more cunning. They increase the dose of levothyroxine because they want to lose weight or take extra lansoprazole before a night out. I defy anyone to find me a person who measures out their dose of antacid. A swig from the bottle is, as we all know, standard practice. 

Now, the corollary to all of this is that the list of pills on a patient’s medical record does not necessarily equate to what they actually take or how they are taken. Some people have several redundant medicines on their drug list that they have not taken for months, or even years. Do not imagine that the fact that they are not taking a tablet and have not done for years means that they stop getting them. They may still get them every month. This is not always the patient’s fault. If they delegate reordering to the local pharmacist, the order may go through “on the nod” every month. There may still be some practices in which prescriptions are pre-printed (or pre-ordered) by the practice, but often it is the patient (or carer) who routinely re-orders everything, including the ones no longer taken; it is easy to just tick every box. Sharing pills is another behaviour that can confound the most assiduous medication reviewer. Husbands and wives share all sorts of things, including their pills.

Before your migraine finally sets in let me just mention the problem with month-long supplies. This is one for which the patient, prescriber and community pharmacist are blameless. It flies in the face of reason that neither the pharmaceutical manufacturers nor prescribing regulators have a unified view as to the number of days in a month. Some drugs come in multiples of 30, and others in multiples of 28. This means that after a couple of years a patient’s drugs get out of kilter, so that the patient needs an extra prescription of those in 28s, but not those in 30s. Unfortunately, patients notice the discrepancy long before two years, and find themselves with a surplus of several weeks supply of two drugs, so they do not order those one month. However, this means that the 30-day pills run out before the end of the next month, so the patient ends up ordering pills twice a month instead of once. In a few months the quantities of drugs the patient holds no longer match at all. 

This always assumes that the quantities on the repeat list match the dosage and frequency on the repeat list, which quite often they don’t.

So, within each person’s drug list there may be:
Drugs that are being taken regularly and appropriately and prescribed monthly in correct quantities for the doses stipulated.Drugs that are intended to be taken as necessary and ordered only when needed.Drugs that are taken at doses or frequency that differs from that stipulated, and are requested at intervals that reflect usage.Drugs that are not being taken at all but are still being requested and prescribed monthly.Drugs that are still on the list but never requested or used.Drugs that are requested monthly, but used intermittently or irregularly.Drugs where the quantity on the drug list does not match the dosage on the drug list.Drugs that run out every month and the patient does not take enough because they are out of kilter.Drugs that run out every month because the patient is taking too many or giving them to their spouse or neighbour or dog.Drugs that are being sold in the pub.Other possibilities that the reader may have experienced or imagine.

Only categories 1 to 3 can truly be said to be eligible for “real” deprescribing. Deprescribing category 4 will save money for the NHS, and avoid the dangerous accumulation of tablets in the patient’s bedside drawer, but will not affect the patient’s health. Deprescribing category 5 will be a paper transaction only. I suggest that stopping these drugs is not deprescribing at all, but **“pseudo-deprescribing”**, because it does not alter what the patient actually receives or swallows. When reading papers studying deprescribing, it is probably sensible to check whether the supposed benefits take account of this phenomenon.

Categories 6 to 11 certainly benefit from identifying and addressing, but do not, I argue, count as deprescribing.

## 8. Sneak Repeat Syndrome—Repeats Acquired without Authorisation

While the essence of medication review (as with all clinical activity) should focus on the patient, it would be remiss not to consider the systems and behaviours whereby people acquire their repeat medications. One common system-based mechanism whereby patients acquire repeat items is the **“sneak repeat”**. The patient (or sometimes the pharmacist on the patient’s behalf) sends a message requesting more pills and the GP repeats the prescription. Now no one would ever admit that patients in *their* practice can obtain repeats continued without formal reconsideration of the concordance, efficacy, safety, and tolerance of the drug, but it happens in other people’s practices, so it probably happens in yours.

It often happens with drugs proposed by hospital specialists. The letter from the consultant suggests or even instructs the GP to prescribe, so the drug is duly prescribed. Even if the doctor intends to review the medication subsequently, the patient responds deferentially “this was started by Mr. Brownsword” (the patient always call him “Mr.”, even if he is not a surgeon), implying that it is immutable. They will want more even if it gives them adverse effects… and so it becomes a repeat.

The process of deciding whether to make a drug a repeat has been described (by me actually) as **authorisation** [8]. Unfortunately, it has never caught on as a concept and I have been unable to discover any subsequent literature on the subject. As Richard Asher pointed out many decades ago, a problem without a proper name produces no literature at all, even when it is a commonly experienced phenomenon. Asher’s use of colourful names for syndromes (such as “Munchausen’s syndrome”) led to a flurry of reports and publications [9]. Therefore, I rename the phenomenon of the repeat that is unauthorised as **sneak repeat syndrome**, in the hope reports will pop up all over the world from others who have recognised the phenomenon but did not know what to call it. There is no doubt that drugs do manage to sneak on to patients’ repeat lists without any evidence in the records that a prescriber had formally considered whether to continue it.

Sneak repeats should be prevented rather than treated. It should not be possible for a drug to get on to a patient’s repeat list without a clinical review to check that it is being taken properly, is having its desired effect, is not causing adverse effects, and (of course) needs to be continued long-term. The difficulty is that it is too easy. If you consider what ought to happen, there are several potential risks in the process. Let us consider a patient newly started on a statin. If you issue a one-off prescription, there is a risk that the patient will not come back for a repeat in a month’s time, even if you ask them to. The next time you see them, perhaps years later, they might be experiencing the central chest pain you are trying to prevent. If you put it on repeat, on the other hand, they may continue to request repeats without anyone checking whether the pills are OK. Ideally, prescribing software should allow for an intermediate status of provisional repeat (perhaps with a high visibility background colour), and should also alert the practice each month to patients on provisional repeats who have not been reviewed.

## 9. Is Deprescribing Valid and Appropriate?

Deprescribing is an important intervention, but it is only one possible outcome of the process of reviewing a patient and their medication. To set out only to deprescribe is to sell the patient short and puts the cart before the horse. It starts with a rather mechanistic intended outcome rather than a patient-centred objective.

What we really want to do is to **review patients, their illnesses and their medicines** to ensure that the medicines are necessary, appropriate, effective and not causing (or potentially causing) harm. In a practice that carefully controls its repeats, you could review dozens of patients and not need to deprescribe once. You might end up discussing the patient’s concerns, explaining what the pills are for, improving concordance, adjusting dosages, switching therapy, monitoring biological variables, discussing non-pharmacological management, even referring to another health professional, and, of course, renewing the patient’s review date. You might even un-deprescribe; that is, restart a medicine that has been deprescribed inappropriately. However, to call this complex process deprescribing is to sell it short and oversimplify it.

## 10. The Role of Clinical Pharmacists

Clinical pharmacists have come a long way in the last two decades. They have acquired and honed interpersonal skills in addition to their inherent pharmacological skills. They have learned about people, as well as illnesses and medicines. They have become holistic health professionals. Doctors have never been good at medication review [8], partly because they have more urgent priorities, but partly because they tend to paint with a broader brush than pharmacists. Pharmacists love detail, and are never happier than when exploring the intricacies of dosage and patients’ views and experiences with their medicines. Stopping pills is only one possible outcome of this complex professional process. Let us not oversimplify it, or else the role of a clinical pharmacist will become easy picking for outsourcing companies, who will appoint undertrained assistants to do the job at half their salaries, using half-baked computer algorithms and ignoring the real issues of long term treatment. What is worse, they will highlight the reduction in the number and cost of pills prescribed as evidence of their own virtuosity, without ever having explored the real issues of the outcome and control of ongoing treatment.

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
