# Peer review of "Deprescribing: Fashion Accessory or Fig Leaf?"

_pharmacy, 2019, doi:10.3390/pharmacy7020049_

Round 1
Reviewer 1 Report
The paper was an interesting discussion of the need for deprescribing medications. This topic is very interesting as many clinicians need to be considering this a high priority.
I have a few comments to be addressed or thought through:
Line 9, second sentence: Starting with thinking prescribers- this is odd- I would consider rewording this
Line 12- do prescribers really benefit from signing less prescriptions- this seems like a strange reason to deprescribe
The second paragraph is very confusing- I would suggest reworking this area
Line 41- some of these medications are not always intended for long term use- I would reconsider the choices here
Line 46- I would consider a different class over antidepressants- this may be used long term not just when the disorder has resolved, depression and anxiety are life long illnesses.
Line 65- I would be careful about saying stopping benzodiazepines- it may not seem obvious to everyone the need to taper this class of medications
Line 149- I would choose another word besides wind
Author Response
Dear Reviewer 1,
Thank you for your interesting and carefully considered thoughts and suggestions.
1. Line 9 “Thinking prescribers […].” I’m glad you raised this question. It is intentional.You (and other readers) are meant to be brought to a halt by this phrase. You are meant to be puzzled. ‘What on Earth does he mean by this phrase? Doesn’t all prescribing require thought? You have to make a diagnosis, consider the risks and natural history of the disease, and then consider whether and what treatment is appropriate, taking into account efficacy, possible adverse effects, acceptability,cost, alternative strategies, patient expectation, guidelines etc. etc. Isn’t this thought? Well of course it is, but it’s not the sort of thought I mean. I am really talking about strategic thinking, such as what prescriptions are for, and is my prescribing safe, and can I be sure that my colleague has considered all the issues when initiating this medicine, and are those considerations just as valid now as they were when the drug was initiated? I could, of course, have put in a paragraph in this ilk… but that would be spoonfeeding the reader, and I really want the reader to stop and think about what he/she is reading. The whole of this article is meant to be provocative. It is not meant to be comfortable reading or bland. If the editor would wish me to do so, I am willing to change the phrase to “Thinking long term prescribers”, but I think this detracts from the impact of the phrase. (Interestingly Reviewer 3 actually uses “thoughtful” to mean the same as my “thinking”… but “thoughtful” is the wrong word, having implications of consideration for the wellbeing of others, solicitousness, considerateness, which isn’t the meaning intended by me or, for that matter, him/her.)
2. Line 12 Do prescribers benefit from signing fewer prescriptions? I am surprised by the reviewer’s comments about the benefits of having fewer prescriptions to sign. The huge pile (formerly paper, but now electronic) of prescriptions to be signed at the end of morning surgery is a chore that prescribers (especially GPs) hate for various reasons:
a. It takes a sizeable chunk of the morning to sign them off.
b. It is boring!
c. It involves risk.
d. There is always the haunting feeling that one is reliant on the professionalism and attention to details of one’s colleagues (and oneself) that the prescriptions approved are appropriate, effective, safe etc. There is simply not time to check back every item.
There is therefore a feeling of mixed unease and guilt. Having fewer to sign reduces the time spent doing them and the time spent worrying about them. And if you stop one drug successfully, you may reduce your workload by many items over the course of a patient lifetime. In fact, I would argue that stopping a drug seldom makes much difference to the cost of medicines, because most repeat drugs are cheap. But the reduction in professional time is tangible and much more valuable as an opportunity benefit. I did not spell this out in the article, as before, because I want to sound a little outrageous so as to provoke thought.
3. I have rewritten para 2.
4. Line 41. I think that it is extremely unusual to prescribe hypotensives, statins or antiepileptics for short term use. I stand by my choices. (They are sometimes stopped or changed because of ineffectiveness or adverse effects, but that is not the issue).
5. Line 46. I agree that antidepressants are often taken for many years, and sometimes for life. I do not agree that depression and anxiety are always life long illnesses, though they can be. But it is still true to say that the only way of telling whether an antidepressant can be stopped is to stop it and see what happens. This is never an easy option, of course, but roughly a third of patients whose depression has been successfully treated can withdraw their medication and have no further depressive illness. A further third will not relapse immediately, but will have another depressive illness at a later date. But the remaining third will relapse rapidly, and some (but not all) of these will need to stay on medication lifelong. [Ref: Geddes JR, Carney SM, Davies C, Furukawa TA, Kupfer DJ, Frank E, et al. Relapse prevention with antidepressant drug treatment in depressive disorders: a systematic review. Lancet. 2003;361(9358):653-61]. Now the question of whether patients should be encouraged to stop antidepressants is an interesting issue, and depends whether the prescriber is a dove or a hawk. As for patients themselves, many stop them without telling us, some ask if they can stop them, some wait for the prescriber to raise the issue and some are fearful of stopping. As with all therapeutic decisions, the best way is generally to discuss the options with the patient and come to a mutually acceptable decision. It is always important to remember that the illness belongs to the patient, and involving the patient in any therapeutic decision is essential.
6. Line 65. Point taken. I have added a qualifying sentence.
7. “Wind”I think the reviewer is being oversensitive. And it doesn’t really matter whether the actual symptom is bloating, burping or flatus – I leave that to the reader’s imagination. Patients use the same word for each or all of these meanings.

Reviewer 2 Report
The term de-prescribing has become very trendy recently but like most trends, there is a risk in focusing on style over substance. I could not agree more with the author's conclusions about the dangers of a simplistic, mechanistic, algorithmic, and, dare I say it, "robotic" approach to deprescribing that completely displaces the patient and their preferences from the centre of the process.
The anecdotes and examples resonate with me, as no doubt they will with all thoughtful GPs and pharmacists. The author makes many pertinent points that demonstrate his familiarity with the problems associated with deprescribing in practice.
Weaknesses are that the author clearly has a strong opinion on this and is not really attempting to present a balanced review of the evidence.
Strengths are that he writes very well, clearly has a lot of practical experience in the field and understands how patients and doctors actually behave in the real world as opposed to in clinical trials.
Author Response
Dear Reviewer 2,
You have taken my article as it is intended to be read, and I am grateful for this. It is intended to be provocative and I hope it incites correspondence and discussion. Yes, I have a strong opinion and I would encourage professional colleagues to form opinions and to express them unequivocally. The literature contains lots of balanced views, and a lot more that is so bland that no view comes across. That is not how progress is made.
Reviewer 3 Report
I was glad to see an editorial on deprescribing that discussed varied sides of the process.
A few suggestions:
Lines 20-1, "But hard evidence of value is still patchy and equivocal." While this is true, there are several ongoing studies that will help bolster the evidence base. We are conducting 2 large RCTs to determine the effect of a deprescribing algorithm on patient-centered outcomes. The protocol was just published: Vasilevskis EE et al. BMC Health Serv Res. 2019 Mar 14;19(1):165.
Taking the patient's preferences into account must also be considered when discussing deprescribing.
There is a body of knowledge around frameworks of deprescribing, including patient and provider barriers and facilitators. At least referencing these, if not commenting on them, would bolster the editorial.
Please use the generic name for Gaviscon.
Paragraph "Deprescribing plus": I would add that there are several other possible measurements of success including decreases in patient-reported outcomes and adverse drug events.
Line 260-1: Please mention the importance of taking a best possible medication history (BPMH) in order to do a thorough clinical review. If the history is incorrect, the review will be inaccurate.
The end of the editorial would benefit from a short summary.
Author Response
Dear Reviewer 3,
I am grateful for your comments and I am interested to hear about your RCTs. I shall be looking out for the results of them. But it is still true to say (and you acknowledge this) that evidence of value is patchy and equivocal. Indeed, your own clinical trial may not show benefit. Otherwise there would be no point in conducting it. My own experience shows how difficult it is to measure meaningful benefit in behavioural research.
My article is not intended to be an Editorial, but an opinion piece, and for that reason it is written slightly provocatively and not in a balanced way. It is intended to provoke discussion and argument, as it has with my three reviewers, each of whose perspective is different.
I have emphasised the need for patient comfort (and carer comfort) with the process, have emphasised the interpersonal and consultation dimension and have suggested how the review process needs to be structured. (New para “The manner and manners of prescribing”). I have also emphasised that not deprescribing is a valid outcome. I have strengthened this by emphasising the need for a unified strategic approach. I hope this also addresses the frameworks issue, though I have not quoted references.
I have changed “Gaviscon” to “antacid”… though I don’t think it reads as well!
I have explored medication history in the above new paragraph, and raised the issue of reconciliation.
Finally, I have resisted the pressure to write a summary. Indeed I intentionally made my abstract annoyingly short and slightly outrageous. It is intended to lead readers in rather than letting them get away with reading a bland summary.
